# Memory Efficient Block Coordinate Descent Method for Forward-Only Second-Order Finetuning of LLM Models

## Abstract

Fine-tuning large language models (LLMs) for specific downstream tasks has traditionally relied on memory-intensive optimizers using classical backpropagation, which demands substantial memory to store model states for gradient computation, motivating the development of memory-efficient zeroth-order optimizers that operate in a forward-only manner. However, the slower convergence of the zeroth-order optimizer remains a challenge, which recent research addresses by incorporating Hessian information to accelerate training, although storing even the diagonal Hessian requires memory equivalent to that of the model weights, leading to significant memory usage. To mitigate this problem, we propose a novel approach that integrates the block coordinate descent (BCD) method with a Hessian-informed zeroth-order optimizer, allowing us to treat model layers as separate blocks and update only a subset of layers per training iteration, thereby reducing memory requirements and accelerating convergence. Specifically, at each iteration, an active block of layers is selected according to the chosen BCD rule, such as ascending order, and their weights are updated while the other layers remain fixed, with diagonal Hessian information stored and updated exclusively for the active layers. For fine-tuning foundation models of medium size (OPT-1.3B and LLaMA-2-7B), our method achieves up to 39% memory reduction compared to existing Hessian-informed zeroth-order methods, while preserving baseline accuracy and memory usage to zeroth-order methods across various tasks, offering a memory-efficient alternative method for LLMs fine-tuning, especially on memory-constrained devices.

## 1. Introduction

Fine-tuning transformer-based large models is an essential step in adapting pre-trained models to specific downstream tasks and further improving performance (Raffel et al., 2020). This process also allows the model to continue training on lower-end devices compared to those used for pre-training, thus improving accessibility and reducing the training cost. For this intent, parameter-efficient fine-tuning (PEFT) techniques, such as LoRA (Hu et al., 2021), have been proposed to enable fine-tuning on consumer-level GPUs or even edge devices, providing significant economic and practical benefits. Typically, fine-tuning employs traditional optimizers like SGD or Adam (Kingma, 2014), which use backpropagation to update model weights. This process requires storing parameters, gradients, activations, and possibly other optimizer states, significantly increasing memory requirements (Lv et al., 2023b;a; Rajbhandari et al., 2020). As model sizes have increased and larger batch sizes are employed for training, the memory demands of traditional optimizers have become a significant bottleneck for devices with limited memory resources, even when using existing PEFT methods (Cai et al., 2020). Our work aims to address this challenge by exploring memory-efficient techniques further to reduce the memory overhead during fine-tuning on low-end devices.

To tackle the memory inefficiency issue, recent advancements have explored the use of zeroth-order optimizers such as MeZO (Malladi et al., 2023) that estimate the gradients with only forward passes, which eliminates the need for backpropagation, thereby significantly reducing memory consumption by avoiding the storage of intermediate optimizer states. Though memory-efficient, the slower convergence rates of zeroth-order optimizers have limited their practical utility. To accelerate convergence, researchers have incorporated second-order information, such as diagonal Hessian approximations as proposed in HiZOO (Zhao et al., 2024b), into the optimization process. However, this solution comes at the cost of memory overhead, as storing even diagonal Hessian values introduces substantial memory cost comparable to the storage required for the model weights themselves, thereby negating the original memory-saving intent of applying zeroth-order optimization.

[1]Anonymous Institution, Anonymous City, Anonymous Region, Anonymous Country. Correspondence to: Anonymous Author <anon.email@domain.com>.

Preliminary work. Under review by the International Conference on Machine Learning (ICML). Do not distribute.

*Figure 1.* Illustration of MeZO, HiZOO and our proposed training pipeline.

To efficiently and effectively utilize second-order information, we consider the traditional block coordinate descent (BCD) method, which solves optimization problems successively along coordinate directions, and propose a block coordinate descent Newton method (BCD-Newton) to tackle our challenge. Inspired by recent advances in applying BCD with Adam (Kingma, 2014) and AdamW Loshchilov (2017) optimizations for training large language models (LLMs) (Pan et al., 2024; Luo et al., 2024), we introduce a layer-wise block coordinate descent scheme to optimize memory usage, treating model layers as independent blocks and selectively activates a subset of layers during each iteration. In practice, block selection is guided by various BCD rules, including ordered selection and block-wise importance sampling, to achieve optimal training performance. This approach significantly reduces the memory required to store Hessian information. In our optimization step, beyond the basic zeroth-order method with two forward passes, a three-step forward pass is employed to incorporate second-order updates, thereby facilitating faster convergence. The second-order term is stored as a diagonal Hessian estimate matrix, sized according to the active block of selected transformer layers, and is updated throughout the training process. Thus, we propose a novel optimizer that addresses the memory-convergence trade-off inherent in Hessian-informed zeroth-order optimization by integrating BCD techniques, while simultaneously improving convergence rates.

Through extensive experiments on a single RTX 4090 or RTX A6000 GPU, we demonstrate that our method enhances training efficiency and memory management while fine-tuning foundation models, including OPT-1.3B (Zhang et al., 2022b) and LLaMA-2-7B (Touvron et al., 2023). As a BCD zeroth-order Newton method, it empirically delivers superior convergence speed and accuracy compared to MeZO. Additionally, compared to the HiZOO baseline, our approach achieves approximately a 50% speedup and a 40% reduction in memory usage with comparable baseline accuracy across multiple GLUE (Wang, 2018) and SuperGLUE (Wang et al., 2019) tasks. These improvements make our method particularly well-suited for fine-tuning large models on devices with limited memory, expanding the accessibility of large language models in real-world applications.

In summary, our main contributions are three-fold:

- We propose a novel block coordinate descent fine-tuning pipeline that integrates the previous Hessian-informed zeroth-order optimizer, reducing the memory overhead to make the method a practical and convergence-enhanced alternative to MeZO.

- We design improved block coordinate descent schemes that reduce the compute and memory cost of the Hessian-informed forward-only optimizer. By adaptively updating weights across block coordinates of the model layers, this method manages blockwise updates efficiently and reduces memory and computational costs.

- We conduct experiments on fine-tuning OPT-1.3B and LLaMA-2-7B, demonstrating that our method reduces training memory by over 39% without a loss in accuracy compared to the full diagonal Hessian baseline.

## 2. Related Work

**First and second order Optimization for LLMs.** Traditional first-order optimizers, such as SGD, AdaGrad (Duchi et al., 2011), and RMSProp (Tieleman et al., 2012), are foundational tools in deep learning. Adam (Kingma, 2014), with its adaptive moment estimates for faster convergence, and its variant AdamW (Loshchilov, 2017), which modifies

the weight decay term to improve generalization, have become the dominant optimizers for fine-tuning large language models (LLMs). Second-order optimization methods incorporating Hessian information, such as K-FAC (Martens & Grosse, 2015), EVA (Zhang et al., 2022a), Adahessian (Yao et al., 2021), and Sophia (Liu et al., 2023), have been explored to further accelerate convergence. However, estimating the Hessian is computationally and memory intensive, particularly with the growing size of LLMs, which makes these second-order methods less practical for fine-tuning on devices with limited memory resources.

**Zeroth-order (ZO) Optimization.** A classical zeroth-order optimization method, SPSA (Spall, 1992), with its corresponding SGD variant, ZO-SGD, estimates the gradient using two forward passes before and after parameter perturbation. Recently, MeZO (Malladi et al., 2023) adapted ZO-SGD by incorporating the random number generator, enabling an in-place implementation significantly reducing memory usage for storing random vectors during training. Based on MeZO, recent work explores its variants like incorporated sparsity for memory efficiency (Guo et al., 2024). Additionally, Zhang et al. (2024) conducted a benchmark study to analyze and enhance zeroth-order fine-tuning methods. However, the convergence performance of ZO methods often falls behind that of first-order methods. To improve convergence, HiZOO (Zhao et al., 2024b) proposed to utilize Hessian information through diagonal Hessian estimation. Beyond these approaches, several other gradient-free methods have been proposed, such as using evolutionary algorithms for gradient-free optimization (Sun et al., 2022b;a).

**Memory-efficient Fine-tuning for LLMs.** Numerous algorithms have been developed to reduce memory costs for training LLMs. Based on backpropagation, practical techniques such as gradient checkpointing (Chen et al., 2016) recompute gradients, FlashAttention (Dao et al., 2022) employs tiling and recomputation to leverage cache for improved efficiency, and the ZeRO optimizers (Rajbhandari et al., 2020; Ren et al., 2021) enable offloading to manage memory usage effectively. Additionally, researchers have utilized compression and quantization methods to approximate gradients, activations, and other optimizer states, enhancing training performance (Jiang et al., 2022; Li et al., 2024). On another front, methods like LOMO (Lv et al., 2023b;a) fuse gradient updates to accelerate training. One notable approach to fine-tuning is parameter-efficient fine-tuning (PEFT) methods, which includes techniques such as Adapters (LoRA) (Hu et al., 2021; Houlsby et al., 2019), prompt tuning (Lester et al., 2021), and selective methods like bias-only fine-tuning (Zaken et al., 2021) and layer-wise freezing (Brock et al., 2017). In addition, Zhao et al. (2024a) recently introduced GaLore which reduces memory costs by projecting gradients into a low-rank compact space.

**Block Coordinate Descent (BCD) methods for LLM Optimization.** In BCD, the optimization objective is minimized successively along coordinate directions. When applied to LLM fine-tuning, this approach can be seen as a branch of selective methods in parameter-efficient fine-tuning. The recently proposed BAdam (Luo et al., 2024) showcases the effectiveness of combining block coordinate descent with Adam. Similarly, LiSA (Pan et al., 2024) improves performance by selectively updating transformer layers with AdamW optimizer, outperforming LoRA across tasks on LLaMA-2.

Overall, our method offers a complementary optimizer-based solution that can be combined with techniques like compression and system-level approaches to improve memory efficiency. Amid the rapid advancements in efficient training for LLMs and other foundation models, the most closely related works to ours are HiZOO and BAdam. However, our approach distinguishes itself by addressing the memory overhead of these methods in two key ways: first, by eliminating the need for backpropagation through zeroth-order optimization, and second, by reducing the memory cost of Hessian-informed methods through block coordinate descent. Unlike PEFT methods, our approach enables full parameter fine-tuning, which has been demonstrated to yield superior performance in various tasks (Ding et al., 2022).

## 3. Revisiting Memory Cost: A BCD Approach

In this section, we provide a brief overview of how zeroth-order (ZO) and Hessian-informed ZO optimizer methods work by introducing the core concepts of MeZO (Malladi et al., 2023) and HiZOO (Zhao et al., 2024b). Next, we introduce block coordinate descent (BCD) methods such as BAdam (Luo et al., 2024). To ensure consistency, we have adapted the definitions from these works. Finally, we reconsider the memory consumption of these methods, and propose our BCD-integrated Newton method optimizer.

### 3.1. Preliminaries of Zeroth-order Optimizers

#### 3.1.1. SPSA, ZO-SGD, AND MEZO

Let $\mathcal{L}(\boldsymbol{\theta}; \mathcal{B})$ represent the loss function for training the model with parameters $\boldsymbol{\theta} \in \mathbb{R}^d$ on the minibatch $\mathcal{B}$, omitting the $\mathcal{B}$ for simplicity. The SPSA algorithm (Spall, 1992) perturbs the model using $\boldsymbol{z} \in \mathbb{R}^d$, sampled from $\mathcal{N}(0, \mathbf{I}_d)$, and estimates the gradient on the minibatch as follows:

$$\hat{\nabla}\mathcal{L}(\boldsymbol{\theta}) = \frac{\mathcal{L}(\boldsymbol{\theta} + \mu\boldsymbol{z}) - \mathcal{L}(\boldsymbol{\theta} - \mu\boldsymbol{z})}{2\mu}\boldsymbol{z} \approx \boldsymbol{z}\boldsymbol{z}^\top \nabla\mathcal{L}(\boldsymbol{\theta}) \quad (1)$$

where $\mu$ is the perturbation scale.

The corresponding SPSA optimizer, ZO-SGD, employs two forward passes to estimate the gradients. With learning rate $\eta$, ZO-SGD updates the parameters as $\boldsymbol{\theta}_{t+1} =$

$\boldsymbol{\theta}_t - \eta\hat{\nabla}\mathcal{L}(\boldsymbol{\theta}; \mathcal{B}_t)$. In this vanilla algorithm, the sampled vector $\boldsymbol{z}$ requires memory equivalent to that of the perturbed weights, resulting in a memory cost that is double the cost of inference. In contrast, MeZO (Malladi et al., 2023) introduces an in-place implementation using a random number generator. Only a random seed $s$ needs to be sampled and stored at each step, allowing the generator to be reset by $s$ to regenerate the vector $\boldsymbol{z}$. This approach eliminates the need to save the vector, reducing the memory cost to match that of inference.

### 3.1.2. HiZOO

To harness second-order information through MeZO for enhanced convergence rates, Zhao et al. (2024b) introduce HiZOO, utilizing a diagonal Hessian-based preconditioner that adjusts the update sizes of parameters based on their curvature. By estimating and storing only the diagonal Hessian, HiZOO requires $\mathcal{O}(d)$ memory, significantly less than the $\mathcal{O}(d^2)$ needed for the full Hessian matrix.

Let $\boldsymbol{\Sigma}$ denote the estimated inverse Hessian matrix, approximating the diagonal Hessian as a positive definite matrix, with $\boldsymbol{\Sigma}^{-1} \approx \nabla^2\mathcal{L}(\boldsymbol{\theta})$. Define $\boldsymbol{\Sigma}_t$ as the estimated Hessian inverse at training step $t$, initialized as $\boldsymbol{\Sigma}_0 = \mathbf{I}_d$. Storing $\boldsymbol{\Sigma}_t$ incurs a memory cost of $\mathcal{O}(d)$, and it is updated at each step. In addition, to mitigate noise in the computation, an exponential moving average (EMA) is employed, leading to the following update rule for the diagonal Hessian estimate:

$$\boldsymbol{\Sigma}_{t+1}^{-1} = (1 - \alpha_t)\boldsymbol{\Sigma}_t^{-1} + \alpha_t |\boldsymbol{\Sigma}_t|, \tag{2}$$

where $\alpha_t$ is a smooth scale, and $|\boldsymbol{\Sigma}_t|$ ensures that all entries of $\boldsymbol{\Sigma}_t$ remain non-negative.

HiZOO approximates the diagonal Hessian using three forward passes to compute $\mathcal{L}(\boldsymbol{\theta} + \mu\boldsymbol{\Sigma}^{1/2}\boldsymbol{z})$, $\mathcal{L}(\boldsymbol{\theta} - \mu\boldsymbol{\Sigma}^{1/2}\boldsymbol{z})$, and $\mathcal{L}(\boldsymbol{\theta})$. By applying Taylor's expansion, they obtain that:

$$\mathcal{L}(\boldsymbol{\theta} \pm \mu\boldsymbol{\Sigma}^{1/2}\boldsymbol{z}) = \mathcal{L}(\boldsymbol{\theta}) \pm \mu\langle\nabla\mathcal{L}(\boldsymbol{\theta}), \boldsymbol{\Sigma}^{1/2}\boldsymbol{z}\rangle$$
$$+ \frac{\mu^2}{2}\boldsymbol{z}^\top\boldsymbol{\Sigma}^{1/2}\nabla^2\mathcal{L}(\boldsymbol{\theta})\boldsymbol{\Sigma}^{1/2}\boldsymbol{z} + \mathcal{O}(\mu^3), \tag{3}$$

the difference $\Delta\mathcal{L}$ is then calculated as:

$$\Delta\mathcal{L} = \mathcal{L}(\boldsymbol{\theta} + \mu\boldsymbol{\Sigma}^{1/2}\boldsymbol{z}) + \mathcal{L}(\boldsymbol{\theta} - \mu\boldsymbol{\Sigma}^{1/2}\boldsymbol{z}) - 2\mathcal{L}(\boldsymbol{\theta})$$
$$= \mu^2\boldsymbol{z}^\top\boldsymbol{\Sigma}^{1/2}\nabla^2\mathcal{L}(\boldsymbol{\theta})\boldsymbol{\Sigma}^{1/2}\boldsymbol{z} + \mathcal{O}(\mu^3).$$

Based on Ye (2023), the following term equals $\nabla^2\mathcal{L}(\boldsymbol{\theta})$,

$$\frac{1}{2} \cdot \mathbb{E}_{\boldsymbol{z}}(\boldsymbol{z}^\top\boldsymbol{\Sigma}^{1/2}\nabla^2\mathcal{L}(\boldsymbol{\theta})\boldsymbol{\Sigma}^{1/2}\boldsymbol{z} \cdot (\boldsymbol{\Sigma}^{-1/2}\boldsymbol{z}\boldsymbol{z}^\top\boldsymbol{\Sigma}^{-1/2} - \boldsymbol{\Sigma}^{-1})), \tag{4}$$

substitute $\Delta\mathcal{L}$, and they show that:

$$\frac{1}{2}\mathbb{E}\left[\frac{\Delta\mathcal{L}}{\mu^2} \cdot \left(\boldsymbol{\Sigma}^{-1/2}\boldsymbol{z}\boldsymbol{z}^\top\boldsymbol{\Sigma}^{-1/2} - \boldsymbol{\Sigma}^{-1}\right)\right] = \nabla^2\mathcal{L}(\boldsymbol{\theta}) + \mathcal{O}(\mu).$$

Consequently, the estimation of the diagonal Hessian $\nabla^2\mathcal{L}(\boldsymbol{\theta})$ at $\boldsymbol{\theta}$ is:

$$\boldsymbol{\Sigma}_t = \frac{\Delta\mathcal{L}}{2\mu^2}\left(\boldsymbol{\Sigma}_t^{-1/2}\boldsymbol{z}_i\boldsymbol{z}_i^\top\boldsymbol{\Sigma}_t^{-1/2} - \boldsymbol{\Sigma}_t^{-1}\right). \tag{5}$$

In this manner, HiZOO approximates the diagonal entries of $\nabla^2\mathcal{L}(\boldsymbol{\theta})$ by $\boldsymbol{\Sigma}_t$, requiring one more forward pass per step compared with MeZO.

### 3.1.3. BLOCK COORDINATE DESCENT

At each iteration, block coordinate descent (BCD) fixes all other parameters and optimizes the objective function over the selected coordinates, resulting in an optimization problem with reduced dimension. For large language models, a natural block partition is to organize transformer layers in ascending order. Formally, an ordered block partition $\pi = \{\pi_1, \ldots, \pi_i, \ldots, \pi_D\}$ divides the entire model parameters $\boldsymbol{\theta} \in \mathbb{R}^d$ into $D$ blocks, such that $\boldsymbol{\theta} = \{\boldsymbol{\theta}_{\pi_1}, \ldots, \boldsymbol{\theta}_{\pi_i}, \ldots, \boldsymbol{\theta}_{\pi_D}\}$ with $\boldsymbol{\theta}_{\pi_i} \in \mathbb{R}^{d_i}$ and $\sum_{i=1}^{D} d_i = d$. Based on the main idea of BCD, BAdam (Luo et al., 2024) propose to incorporate Adam updates as its inner solver and optimize over only one active block $\boldsymbol{\theta}_{\pi_i}$ at a time while keeping the other inactive blocks fixed. Mathematically, BAdam solves the following subproblem at the $t$-th block-epoch for $i = 1, \ldots, D$ to update the active block $\boldsymbol{\theta}_{\pi_i}$:

$$\boldsymbol{\theta}_{\pi_i}^{t+1} \in \arg\min_{\boldsymbol{\theta}_{\pi_i} \in \mathbb{R}^{d_i}} \mathcal{L}(\boldsymbol{\theta}_{\pi_1}^{t+1}, \ldots, \boldsymbol{\theta}_{\pi_{i-1}}^{t+1}, \boldsymbol{\theta}_{\pi_i}, \boldsymbol{\theta}_{\pi_{i+1}}^t, \ldots, \boldsymbol{\theta}_{\pi_D}^t). \tag{6}$$

This subproblem Equation 6 keeps inactive blocks fixed at their latest values, leading to a significantly lower-dimensional optimization problem compared to $\min_{\boldsymbol{\theta}} \mathcal{L}(\boldsymbol{\theta})$.

### 3.2. Revisiting Memory Cost from the BCD Perspective

**Who consumed my memory?** Second-order methods incorporate full or diagonal Hessian matrix, or its estimation, as a preconditioner to accelerate convergence, but this introduces a significant memory cost of $\mathcal{O}(d)$. For large models such as LLaMA-2-7B (Touvron et al., 2023) with $d = 7$ billion parameters, this requires $2d$ memory in FP16 precision, resulting in approximately over 14GB of memory storage. When combined with the memory required for model parameters, this easily exceeds the capacity of consumer-level devices, undermining MeZO's original goal of achieving memory efficiency. Our experiments further demonstrate that directly applying Hessian-based optimization steps significantly increases memory usage, as shown in Table 1. Even though approaches such as HiZOO offer performance improvements, the considerable memory overhead from storing Hessian information becomes a bottleneck, particularly when fine-tuning large models. This dilemma leads to a situation where the benefits of second-order methods are

*Table 1.* Experiments of actual GPU memory consumption for various algorithms.

| DEVICE | MODEL | SGD | BCD | LORA | MEZO | HIZOO | OURS B-PDF |
|--------|-------|-----|-----|------|------|-------|------------|
| RTX 4090 | OPT-1.3B | 23G | 21G | 11G | 4.4G | 7.5G | 4.6G |
| RTX A6000 | LLAMA-2-7B | 48G | 46G | 40G | 31G | 48G | 32G |

| THEORETICAL AVG MEMORY IN UNITS PARAM+ACTIVATION+GRAD+HESSIAN | SGD $3d$ | BCD OR LORA $d \sim 3d$ | MEZO $d$ | HIZOO $2d$ | OURS B-PDF $d + d/D$ |
|---|---|---|---|---|---|

outweighed by their heavy memory consumption, limiting their practicality in memory-constrained environments. Furthermore, the memory consumption increases with batch size for both first-order and Hessian-based methods, intensifying the memory overhead, as illustrated in Figure 2.

**How to reduce Hessian memory consumption?** To address this memory-convergence trade-off, we propose integrating block coordinate descent (BCD) into the zeroth-order Newton optimization. BCD allows us to partition the model into blocks, optimizing only a subset of layers at each iteration while keeping the rest fixed. This approach dramatically reduces the memory required for storing Hessian information, as it is only computed for the active blocks. For instance, by partitioning the aforementioned LLaMA-2-7B model into $D = 32$ blocks, corresponding to its 32 transformer layers, we reduce the additional memory cost associated with Hessian storage to $\frac{2d}{D}$, bringing it to under 1GB of memory. This significantly improves memory efficiency while preserving the advantages of second-order optimization. Moreover, we further optimize memory usage by applying MeZO to update the embedding and language modeling head layers, avoiding the instability and overhead often associated with second-order methods. Our integration of BCD not only achieves comparable memory usage to MeZO but also leverages the improved convergence rates of Hessian-informed updates.

To validate our analysis, we conducted preliminary experiments (detailed in Section 5) measuring the GPU memory consumption of various optimizers during the fine-tuning of medium-sized language models, specifically OPT-1.3B (Zhang et al., 2022b) on an RTX 4090 (24GB) and LLaMA-2-7B on an RTX A6000 (48GB). As Table 1 and Figure 2 briefly illustrate, HiZOO's incorporation of second-order information increases memory demand by over 70%. Notably, the actual allocated memory includes residual state memory such as temporary buffers and fragments (Rajbhandari et al., 2020), which means the overall memory requirement exceeds that of the parameters alone, resulting in the overall increase short of a full 100%. In contrast, our BCD-integrated method significantly reduces memory consumption, bringing it in line with MeZO while maintaining comparable performance. As we will further demonstrate in Section 5, our proposed B-PDF method achieves comparable accuracy, and offers a practical, memory-efficient alternative to MeZO with the extra benefit of incorporating second-order information.

**Flexibility in BCD Block Selection.** Beyond the natural block partitioning of model layers in ascending order, BCD can be adapted with various strategies such as descending order, random reshuffling, or importance sampling (Luo et al., 2024; Pan et al., 2024). For instance, LiSA (Pan et al., 2024) proposes a layer-wise importance sampling approach, which updates selected layers while keeping others frozen, utilizing AdamW as the optimizer. In this approach, layers are randomly selected based on predefined probability values. In Section 4.1, we will present several BCD methods for block selection. This flexibility allows BCD to be adapted to different optimization scenarios, enhancing the overall training process while maintaining memory efficiency.

## 4. Methodology

### 4.1. BCD-integrated ZO-Newton Optimizer

Motivated by the revisiting of the second-order Hessian memory consumption, we identified a significant bottleneck caused by the storage of diagonal Hessian estimation, which introduced substantial memory overhead, particularly for large models. Ultimately, to address this memory-convergence trade-off, we propose a new method that integrates block coordinate descent (BCD) with a zeroth-order Newton optimizer, termed **B**lock-wise diagonal-Hessian **P**reconditioned Coordinate **D**escent **F**orward-only optimizer (**B-PDF**). Recognizing the layerwise structure of the transformer model, we treat each layer as a block for Hessian-informed zeroth-order optimization. By partitioning the model into blocks and updating only a subset of layers at each iteration, we reduce the Hessian storage requirement while maintaining the convergence benefits of second-order methods. Additionally, we update the embedding and language model (LM) head layers solely through ZO optimization to mitigate the instability and overhead typically associated with second-order methods, resulting in a more memory-efficient and scalable approach for fine-tuning.

The block partitioning is naturally arranged in ascending order, and various BCD algorithms can be employed to determine the active block $\boldsymbol{\theta}_{\pi_i}$. Possible strategies include using ascending order, a layerwise importance sampling scheme based on the mean weight norms of each block $\pi_i$, the Gaussian-Southwell-Diagonal rule (Nutini et al., 2017), or dynamically updated probabilistic lists employing a bandit method. Formally, for the current step $T$, the parameter block $\boldsymbol{\theta}_{\pi_b}$ to update can be selected using several types of

BCD algorithms, in which $\boldsymbol{\theta}_{\pi_b}$ is defined by the rule:

$$
\begin{cases}
\boldsymbol{\theta}_{\pi_i}, i \leftarrow [1, \cdots, D], & \text{ordered / random,} \\
\arg\max_{\boldsymbol{\theta}_{\pi_i}} \frac{1}{T} \sum_{t=1}^{T} \|\boldsymbol{\theta}_{\pi_i}^t\|_2, & \text{mean weight norms,} \\
\arg\max_{\boldsymbol{\theta}_{\pi_i}} \frac{|\Delta\mathcal{L}(\boldsymbol{\theta}_{\pi_i})|^2}{\Sigma_{\pi_i}}, & \text{Gauss-Southwell-Diagonal,} \\
\boldsymbol{\theta}_{\pi_z}, \boldsymbol{z} \sim \boldsymbol{p_z}, & \text{importance sampling / bandit,} \\
\cdots
\end{cases}
$$

We note that while different methods can be effective in their original settings, they vary significantly in terms of memory and computational costs. Further details are provided in the Appendix. In practice, the substantial computation and storage required for updates by importance-score-based methods led us to select the more efficient default ascending order rule, and our experiments empirically demonstrate its performance. Now we present the pseudocode for the proposed algorithm in Algorithm 1.

---

**Algorithm 1** Training Pipeline of the Propose B-PDF.

0: **Input:** parameters $\boldsymbol{\theta} \in \mathbb{R}^d$, loss function $\mathcal{L}$, perturbation scale $\mu$, learning rate $\eta$, smooth scale $\alpha$
0: **for** $t = 1, \ldots, T$ **do**
0:     Select block $\boldsymbol{\theta}_{\pi_b} \in \boldsymbol{\theta}$ according to the BCD rule
0:     **if** a new block is selected **then**
0:         $\boldsymbol{\Sigma} \leftarrow \mathbf{I}_{|\boldsymbol{\theta}_{\pi_b}|}$ {Diagonal Hessian initialization}
0:     **end if**
0:     Freeze other layers
0:     Sample a random seed $s$ {First-time sampling}
0:     **for** $\mu_i = 0, +\mu, -2\mu$ **do**
0:         **for** $\boldsymbol{\theta}_i \in \boldsymbol{\theta}_{\pi_b}$ **do**
0:             Sample $\boldsymbol{z} \sim \mathcal{N}_s(0, \mathbf{I}_{|\boldsymbol{\theta}_i|})$
0:             $\boldsymbol{\theta}_i \leftarrow \boldsymbol{\theta}_i + \mu_i \boldsymbol{\Sigma}_t^{1/2} \boldsymbol{z}$ {In-place perturbation}
0:         **end for**
0:         $\ell_{\text{sign}(\mu_i)} = \mathcal{L}(\boldsymbol{\theta})$
0:     **end for**
0:     $\hat{\boldsymbol{\Sigma}}_t \leftarrow \frac{\Delta\ell}{2\mu^2} \boldsymbol{\Sigma}_{t-1}^{-1/2} \boldsymbol{z}_i \boldsymbol{z}_i^\top \boldsymbol{\Sigma}_{t-1}^{-1/2}$ {Hessian Update}
0:     $\boldsymbol{\Sigma}_t^{-1} \leftarrow (1 - \alpha_t)\boldsymbol{\Sigma}_{t-1}^{-1} + \alpha_t \left|\text{diag}(\hat{\boldsymbol{\Sigma}}_t)\right|$
0:     $\texttt{projected\_grad} \leftarrow (\ell_+ - \ell_-)\boldsymbol{\Sigma}_t^{1/2}/2\mu$
0:     Reset random number generator with seed $s$
0:     **for** $\boldsymbol{\theta}_i \in \boldsymbol{\theta}_{\pi_b}$ **do**
0:         Sample $\boldsymbol{z} \sim \mathcal{N}_s(0, \mathbf{I}_{|\boldsymbol{\theta}_i|})$
0:         $\boldsymbol{\theta}_i \leftarrow \boldsymbol{\theta}_i - \eta_t * \texttt{projected\_grad} * \boldsymbol{z}$
0:     **end for**
0: **end for**=0

---

**Remark 1.** The optimization objective is to minimize the loss function $\mathcal{L}(\boldsymbol{\theta})$ by leveraging diagonal Hessian preconditioning within the memory-efficient framework. During each iteration, after selecting the active blocks for updates, zeroth-order optimization with a diagonal Hessian preconditioner is performed for the chosen layers. The diagonal Hessian estimate will be reinitialized for a newly selected block, and updates for that block will occur over several subsequent iterations. The algorithm applies in-place perturbations to the parameters in three steps with the perturbation scale corresponding to $\mu_i = 0, +\mu, -2\mu$, sampling a normally distributed random vector $\boldsymbol{z}$ to perturb the selected block $\boldsymbol{\theta}_{\pi_b}$. For each perturbation, the loss function $\mathcal{L}(\boldsymbol{\theta})$ is computed to estimate the gradient information. Afterward, the diagonal Hessian is updated based on the difference in the computed losses from the perturbed parameters. The gradient for the selected block is then projected using the updated Hessian, and the weights of the active block are updated accordingly.

**Remark 2.** The proposed algorithm efficiently combines BCD with a zeroth-order Newton method by updating only a subset of model layers per iteration. This approach reduces memory usage by eliminating backpropagation and utilizing block-wise gradient updates, while maintaining convergence speed through the use of diagonal Hessian approximations. The consistent use of random vectors and selective parameter perturbation further enhance the method's memory efficiency.

### 4.2. Convergence Analysis

As a BCD variant of HiZOO (Zhao et al., 2024b), our proposed B-PDF preserves its convergence properties. Since our focus is primarily on the practical analysis and implementation of memory efficiency, this work emphasizes practical solutions over theoretical exploration. Nevertheless, we provide a brief summary adapted from their convergence analysis. Adopt the classical assumptions and update rule $\boldsymbol{\theta}_{t+1} = \boldsymbol{\theta}_t - \eta_t \hat{\nabla}\mathcal{L}_\mu(\boldsymbol{\theta}_t)$ as detailed by Zhao et al. (2024b), with iteration number $T$ and a suitable step size $\eta_t$, we have:

$$
\mathbb{E}[\frac{1}{T} \sum_{t=1}^{T} \|\nabla\mathcal{L}(\boldsymbol{\theta}_t)\|^2] \leq \mathcal{O}(\frac{1}{\sqrt{T}})(\mathcal{L}(\boldsymbol{\theta}_0) - \mathcal{L}^*) + \mathcal{O}\left(\mu^2\right),
\tag{7}
$$

where $\mathcal{L}^*$ denotes the minimization of the function $\mathcal{L}(\boldsymbol{\theta}; \mathcal{B})$. As training progresses, the first term on the right-hand side of the equation gradually diminishes to zero, while the second term remains bounded by the perturbation scale. This establishes that our method converges to a bounded neighbourhood around a stationary point. Moreover, as $T \to \infty$, the method converges to the optimal point, as demonstrated by the equation above. A brief proof adapted from HiZOO (Zhao et al., 2024b) is provided in the Appendix. For further theoretical details, we refer readers to their original work.

## 5. Experiments

In this section, we build on the experimental settings of MeZO (Malladi et al., 2023) and HiZOO (Zhao et al., 2024b) to evaluate our proposed B-PDF method in terms of memory

consumption, runtime, and convergence. Our experimental code builds on their open-source repositories, with the block coordinate descent method integrated. To facilitate implementation and reduce resource requirements, we focus on performance across several GLUE and SuperGLUE tasks, following their approach. All experiments are conducted on either a single RTX 4090 (24GB) or RTX A6000 (48GB) GPU. Specific details regarding the hyperparameter grids and implementations are provided in the appendix.

### 5.1. Experiments on OPT-1.3B

**Settings.** First, we conduct experiments by fine-tuning the OPT-1.3B model on a single RTX 4090 GPU. Following the settings of previous work, we select several GLUE and SuperGLUE tasks to evaluate the performance of our proposed B-PDF method. These NLP tasks include sentence classification and text generation. We note that MeZO highlights the significance of incorporating prompts for optimal performance and is structured accordingly. Therefore, we maintained MeZO's original setup and refrained from introducing additional baselines in our experiments. For the first-order baselines, we include SGD, BCD-based SGD (referred to as BCD in the tables), and LoRA with SGD. For the zeroth-order methods, we compare MeZO, HiZOO, and our proposed B-PDF. The batch size is set to 8 for zeroth-order methods and 2 for first-order methods to prevent memory exhaustion. Our primary goal is to demonstrate that B-PDF reduces HiZOO's memory consumption while maintaining speed and accuracy.

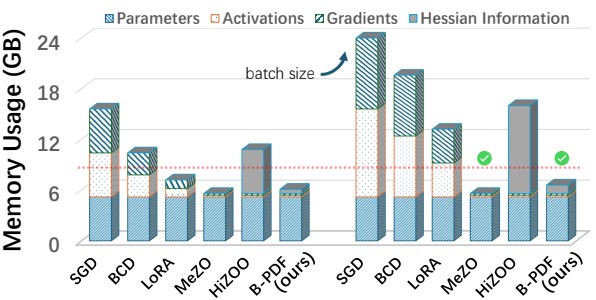

*Figure 2.* Illustration of average GPU memory consumption for fine-tuning the OPT-1.3B model using different methods, with `batchsize = {1, 2}`. As the batch size increases, our proposed B-PDF and MeZO maintain low memory usage, while other methods easily surpass the memory threshold of devices (red dashed line represents an 8GB memory limit on low-end devices).

**Memory Efficiency.** For memory efficiency, B-PDF significantly reduces the memory overhead of incorporating Hessian information while maintaining accuracy. As shown in Tables 1 and 2, our report on average GPU memory usage during experiments demonstrates that B-PDF has a comparable memory cost to MeZO, while offering substantial savings in memory consumption compared to HiZOO and first-order methods such as SGD, BCD-SGD, and LoRA

(`rank=8`). This notable improvement ensures the practical adoption of the proposed method on low-end devices, where memory is a primary bottleneck for training, which is also the original reason why the forward-only approach was developed to save memory down to inference-level requirements. This makes our method a suitable solution for low-memory training environments. In contrast, HiZOO incurs a significant 72% higher memory cost than MeZO, indicating an impractical convergence-memory tradeoff in memory-limited scenarios. Additionally, first-order methods consume even more memory due to the overhead introduced by backpropagation. For instance, BCD-SGD still requires nearly full fine-tuning memory to store activations and gradients for backpropagation. Consequently, due to their substantial memory demands, they are rendered impractical in low-end environments, making faster convergence irrelevant. This further highlights the advantages and rationale of our approach.

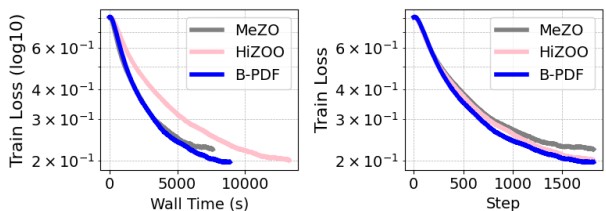

*Figure 3.* Convergence curves of MeZO, HiZOO and proposed B-PDF on SST-2 training OPT-1.3B.

**Convergence Study.** Regarding convergence rate, we present the convergence curve relative to wall-clock time or steps for training on the SST-2 dataset, as illustrated in Figure 3. The results show that while HiZOO converges more effectively than MeZO for 20,000 steps, it requires nearly double the completion time. Conversely, our proposed B-PDF achieves better convergence than MeZO and matches the performance of HiZOO, while maintaining the time efficiency of MeZO. This speedup is attributed to the application of the BCD strategy, which activates only a subset of layers, thereby reducing computational demands. In our experiment, the subset consists of two layers per iteration. As a result, our method benefits from both zeroth-order and Newton methods, thanks to the use of BCD. Furthermore, the results presented in Table 3 and visualized in the appendix demonstrate that our method achieves comparable accuracy to baseline methods across benchmarks. While first-order methods yield superior results, their memory consumption is several times higher than that of zeroth-order methods, making them impractical for low-end environments. In contrast, our method improves memory efficiency while enhancing convergence, outperforming the MeZO baseline and offering a practical, efficient solution for low-end settings. These findings position B-PDF as a memory-efficient optimizer and an effective alternative to MeZO.

*Table 2.* Experiments on OPT-1.3B on SST-2 dataset. The first-order method exceeds the memory limit of low-end devices.

| Method | | | Accuracy | Runtime | | Average Memory | |
|---|---|---|---|---|---|---|---|
| First-Order | Forward+Backward | SGD | 94.3 | 4min 05s | | 22.7 GB | high memory demand |
| | | BCD | 92.4 | 3min 09s | | 20.5 GB | high memory demand |
| | | LoRA | 92.0 | 0min 55s | | 10.6 GB | not full fine-tuning |
| Zeroth-Order | 2×Forward | MeZO | 91.7 | 54min 55s | baseline | **4.4 GB** | baseline |
| | 3×Forward | HiZOO | 91.7 | 99min 44s | + 81.61% | 7.5 GB | + 72.25% |
| | 3×Forward | (**ours**) | **91.9** | **51min 38s** | - 6.98% | 4.6 GB | comparable |

*Table 3.* Experiments on OPT-1.3B across different datasets.

| Task | | SST-2 | RTE | CB | BoolQ | WSC | WIC | SQuAD | Average |
|---|---|---|---|---|---|---|---|---|---|
| Task Type | | ——————-classification—————— | | | | | | generation | |
| First-Order | SGD | 94.3 | 68.6 | 71.4 | 70.0 | 63.5 | 61.4 | 81.6 | 73.0 |
| | BCD | 92.4 | 69.7 | 69.6 | 63.2 | 63.5 | 61.6 | 78.8 | 71.3 |
| | LoRA | 92.4 | 66.4 | 69.6 | 66.8 | 63.5 | 58.5 | 80.5 | 71.1 |
| Zeroth-Order | MeZO | 91.7 | 64.3 | 69.6 | 65.5 | 63.5 | 57.7 | 77.9 | 70.0 |
| | HiZOO | 91.7 | 64.6 | 71.4 | 65.5 | 63.5 | 58.5 | 78.7 | 70.6 |
| | (**ours**) | 91.9 | 65.3 | 69.6 | 65.2 | 63.5 | 57.7 | 77.9 | 70.2 |

*Table 4.* Experiments of fine-tuning LLaMA-2-7B on SST-2 dataset on an RTX A6000 (48 GB).

| Zeroth-Order Method | Accuracy | Average Memory | | First-Order Method | Accuracy | Average Memory | |
|---|---|---|---|---|---|---|---|
| MeZO | 85.2 | **31GB** | baseline | LoRA | 94.8 | 41GB | +32.3% |
| B-PDF | **90.6** | 32GB | + 3.23% | OOM for SGD, BCD, and HiZOO. | | | |

## 5.2. Experiments on LLaMA-2-7B

To further evaluate our proposed method on larger models, we fine-tuned a LLaMA-2-7B model in FP16 precision on an RTX A6000 GPU, using the aforementioned optimization algorithms, as shown in Table 4. Due to the increased model size, both the first-order method and HiZOO encountered out-of-memory (OOM) errors, and B-PDF required longer completion times because of the higher computational cost associated with the larger Hessian estimation. We compared the performance of three methods: LoRA (`rank=8`), MeZO, and our proposed B-PDF, on the SST-2 dataset with `batchsize=1`. The remaining settings were kept consistent with those in Section 5.1. Despite the limited batch size and hardware constraints, which caused an accuracy drop from incomplete convergence, B-PDF still demonstrated performance gains while maintaining comparable memory consumption as a Hessian-informed method, unlike first-order methods draining GPU memory, underscoring its potential in memory-constrained environments.

## 6. Conclusion

In this paper, we propose a novel memory-efficient zeroth-order Newton method that integrates block coordinate descent (BCD) with a diagonal Hessian-preconditioned zeroth-order optimizer for fine-tuning large language models (LLMs). Our approach effectively mitigates the substantial memory overhead commonly associated with second-order methods by employing selective block-wise updates. By combining the BCD technique with the Hessian preconditioner, we achieve significant reductions in memory consumption while preserving competitive accuracy and convergence speed performance. Our extensive experiments on OPT-1.3B and LLaMA-2-7B demonstrate that our method can reduce memory usage by up to 39% compared to existing second-order optimizers while maintaining baseline accuracy across various downstream tasks. Furthermore, our approach exhibits faster wall-clock convergence than conventional zeroth-order methods, making it a practical and scalable solution for fine-tuning large models on resource-constrained devices. Future work will aim to extend this methodology to larger models and more complex tasks, as well as refine the block selection strategies to further enhance both efficiency and performance. In summary, our method provides a promising direction for memory-efficient fine-tuning of LLMs, offering practical advantages, particularly in memory-limited environments.

## 7. Impact Statement

This paper presents work whose goal is to advance the field of efficient training and fine-tuning of foundation models. There are many potential societal consequences of our work, none of which we feel must be specifically highlighted here.

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

## A. Implementation Details

The implementation of the B-PDF function is designed to enhance zero-order optimization (ZOO) by incorporating selective layer-wise updates based on Hessian-informed perturbations.

In the `zo_Hessian_step_update` function (Zhao et al., 2024b), the Hessian matrix is initialized if it does not already exist. This matrix is created by iterating over each trainable parameter of the model and initializing a tensor of ones with the same dimensions as the respective parameter. The estimate Hessian matrix serves as a second-order approximation that is updated during the optimization process.

In our framework, we introduce a layer-specific update mechanism within the optimization function `hizoo_step_update`, which implements a periodic selection of layers, referred to as "hizoo layers." These layers are chosen iteratively every fixed number of steps, with the cycle determined by a step counter.

The layers can be selected either sequentially, in an ordered manner, or via other rules such as Gauss-Southwell quadratic diagonal selection (GSQ) (Nutini et al., 2017), which prioritizes layers based on previous scores.

Following MeZO (Malladi et al., 2023) and HiZOO (Zhao et al., 2024b), we apply a noise-based perturbation to the selected Hessian-informed layers during each iteration using Gaussian noise. The noise is scaled by the square root of the corresponding Hessian matrix and a random vector sampled from a normal distribution. This approach allows the optimization process to focus on specific layers while updating their parameters iteratively.

By controlling the frequency and scope of these updates, we distribute optimization efforts across different parts of the network over time. This can ensure that updates are not applied uniformly but are instead targeted based on layer importance, thereby improving the overall efficiency of the training process.

Additionally, memory management is considered throughout the implementation, as the Hessian matrix is periodically cleared, and GPU memory is freed using `torch.cuda.empty_cache()`, ensuring that the training process remains efficient, even in memory-constrained environments.

In addition, we use `torch.clamp` API to clamp the intermediate results to meet the precision requirements and reduce the instability of second-order methods.

Overall, the B-PDF implementation introduces a structured and targeted optimization approach that leverages layer-wise perturbations to enhance the zero-order optimization process effectively.

## B. Hyperparameter Search

Here, we present the detailed hyperparameter grids used in our experiments, as shown in Table. 5. Empirically, we found that the optimal learning rate for B-PDF is an order of magnitude higher than that for MeZO. Some outlier values in the results may stem from insufficient parameter search or incomplete convergence, likely caused by limited training steps and small batch sizes due to hardware memory constraints.

| Model | Method | Hyperparameters | Values |
|---|---|---|---|
| General Settings in Common | | Learning rate schedule | Linear decay |
| | | Steps | 20000 |
| | | LoRA rank | 8 |
| OPT-1.3B | First-order | Batch size | $\{1, 2\}$ |
| | | Learning rate | $\{1, 3\}$or$\{5, 7\} \times \{1e{-}6, 1e{-}7\}$ |
| | | $\mu$ | $1e{-}3$ |
| | | Weight Decay | 0 |
| OPT-1.3B | Zeroth-order | Batch size | $\{1, 2, 8\}$ |
| | | Learning rate | $\{1, 3\}$or$\{5, 7\} \times \{1e{-}5, 1e{-}6\}$ |
| | | $\mu$ | $1e{-}3$ |
| | | Weight Decay | 0 |
| | | Hessian Smooth Type | Constant $1e{-}9$ |
| | | BCD-Hessian Smooth Type | Constant $1e{-}5$ |
| | | BCD-Update Interval | $\{5, 10\}$ |
| | | BCD-selected layers | $\{1, 2\}$ |
| LLaMA-2-7B | First or Zeroth-order | Batch size | $\{1\}$ |
| | | Learning rate | $\{3\} \times \{1e{-}6, 1e{-}7\}$ |
| | | $\mu$ | $1e{-}3$ |
| | | Weight Decay | 0 |

*Table 5.* The hyperparameter grids used for OPT-1.3B and LLaMA-2-7B experiments.

## C. Additional Visulization Results

Here, we present the bar chart illustrating the test results of OPT-1.3B, as shown in Figure. 4.

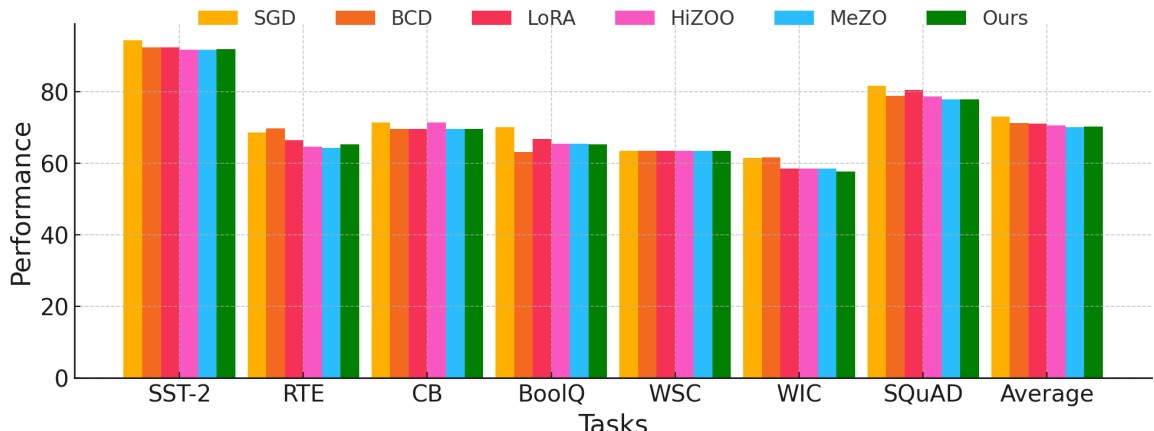

*Figure 4.* Bar chart illustrating the results of training OPT-1.3B with different methods across various benchmarks.

## D. Convergence Analysis

As a block coordinate descent variant of HiZOO (Zhao et al., 2024b), our proposed B-PDF retains the convergence properties of HiZOO. Since our focus is on practical memory reduction rather than theoretical analysis, we offer a brief convergence analysis of our method, adapted from Zhao et al. (2024b), with adjustments made primarily for consistency. For more in-depth theoretical details, we direct readers to their original work.

We adopt several classical assumptions:

**Assumptions.** 1. The objective function $\mathcal{L}(\boldsymbol{\theta}; \mathcal{B})$ is $L_d$-smooth with respect to $\boldsymbol{\theta}_d$, and $L_\infty = \max_d L_d$ ; 2. The stochastic gradient $\nabla\mathcal{L}(\boldsymbol{\theta}; \mathcal{B})$ has $\sigma^2$ variance, i.e. $\mathbb{E}\left[\|\nabla\mathcal{L}(\boldsymbol{\theta}; \mathcal{B}) - \nabla\mathcal{L}(\boldsymbol{\theta})\|^2\right] \le \sigma^2$ ; 3. Each entry of $\boldsymbol{\Sigma}_t$ lies in the range $[\beta_\ell, \beta_u]$ with $0 < \beta_\ell \le \beta_u$.

The the descent direction $\hat{\nabla}\mathcal{L}_\mu(\boldsymbol{\theta}_t)$ defined as:

$$\hat{\nabla}\mathcal{L}_\mu = \sum_{i=1}^{\pi_b} \frac{\mathcal{L}(\boldsymbol{\theta}_t + \mu\boldsymbol{\Sigma}_t^{1/2}\boldsymbol{z}_i; \mathcal{B}) - \mathcal{L}(\boldsymbol{\theta}_t - \mu\boldsymbol{\Sigma}_t^{1/2}\boldsymbol{z}_i; \mathcal{B})}{2b\mu}\boldsymbol{\Sigma}_t^{1/2}\boldsymbol{z}_i. \tag{8}$$

and update rule is $\boldsymbol{\theta}_{t+1} = \boldsymbol{\theta}_t - \eta_t\hat{\nabla}\mathcal{L}_\mu(\boldsymbol{\theta}_t)$.

*Proof.* By the update rule of $\boldsymbol{\theta}_t$ and above assumptions, we have

$$\mathbb{E}\left[\mathcal{L}(\boldsymbol{\theta}_{t+1})\right] - \mathbb{E}\left[\mathcal{L}(\boldsymbol{\theta}_t)\right]$$

$$\le -\eta_t\mathbb{E}\left[\langle\nabla\mathcal{L}(\boldsymbol{\theta}_t), \hat{\nabla}\mathcal{L}_\mu(\boldsymbol{\theta}_t)\rangle\right] + \frac{L_\infty\eta_t^2}{2}\mathbb{E}\left[\|\hat{\nabla}\mathcal{L}_\mu(\boldsymbol{\theta}_t)\|^2\right]$$

$$\le -\eta_t\|\nabla\mathcal{L}(\boldsymbol{\theta}_t)\|_{\boldsymbol{\Sigma}_t}^2 + \eta_t\mathcal{O}\left(\mu\|\nabla\mathcal{L}(\boldsymbol{\theta}_t)\|\right)$$
$$+ 2\eta_t^2 L_\infty\left(\text{tr}(\boldsymbol{\Sigma}_t) + \beta_u\right)\|\nabla\mathcal{L}(\boldsymbol{\theta}_t)\|_{\boldsymbol{\Sigma}_t}^2$$
$$+ 2\eta_t^2 L_\infty\left(\text{tr}(\boldsymbol{\Sigma}_t) + \beta_u\right)\sigma^2 + \mathcal{O}(\mu^2)$$

$$\le -\frac{\eta_t}{2}\|\nabla\mathcal{L}(\boldsymbol{\theta}_t)\|_{\boldsymbol{\Sigma}_t}^2 + 2\eta_t^2 L_\infty\left(\text{tr}(\boldsymbol{\Sigma}_t) + \beta_u\right)\|\nabla\mathcal{L}(\boldsymbol{\theta}_t)\|_{\boldsymbol{\Sigma}_t}^2$$
$$+ 2\eta_t^2 L_\infty\left(\text{tr}(\boldsymbol{\Sigma}_t) + \beta_u\right)\sigma^2 + \mathcal{O}(\mu^2)$$

$$= -\frac{\eta_t}{2}\left(1 - 4\eta_t L(\text{tr}(\boldsymbol{\Sigma}_t) + \beta_u)\right)\|\nabla\mathcal{L}(\boldsymbol{\theta}_t)\|_{\boldsymbol{\Sigma}_t}^2$$
$$+ 2\eta_t^2 L_\infty\left(\text{tr}(\boldsymbol{\Sigma}_t) + \beta_u\right)\sigma^2 + \mathcal{O}(\mu^2)$$

$$\le -\frac{\eta_t}{4}\|\nabla\mathcal{L}(\boldsymbol{\theta}_t)\|_{\boldsymbol{\Sigma}_t}^2 + 2\eta_t^2 L_\infty\left(\text{tr}(\boldsymbol{\Sigma}_t) + \beta_u\right)\sigma^2 + \mathcal{O}(\mu^2),$$

where the second inequality is derived from the following lemma (Zhao et al., 2024b):

$$\mathbb{E}\left[\hat{\nabla}\mathcal{L}_\mu(\boldsymbol{\theta}_t)\right] = \boldsymbol{\Sigma}_t\nabla\mathcal{L}(\boldsymbol{\theta}_t) + \mathcal{O}(\mu)$$
$$\mathbb{E}\left[\|\hat{\nabla}\mathcal{L}_\mu(\boldsymbol{\theta}_t)\|^2\right] \le 4\left(\text{tr}(\boldsymbol{\Sigma}_t) + \beta_u\right)\|\nabla\mathcal{L}(\boldsymbol{\theta}_t)\|_{\boldsymbol{\Sigma}_t}^2 + 4\beta_u\left(\text{tr}(\boldsymbol{\Sigma}_t) + \beta_u\right)\boldsymbol{\Sigma}^2 + \mathcal{O}(\mu^2).$$

By rearranging and summing over T iterations, we have:

$$\mathbb{E}\left[\frac{1}{T}\sum_{t=1}^{T}\|\nabla\mathcal{L}(\boldsymbol{\theta}_t)\|^2\right] \le \frac{1}{T\beta_\ell}\sum_{t=1}^{T}\|\nabla\mathcal{L}(\boldsymbol{\theta}_t)\|_{\boldsymbol{\Sigma}_t}^2$$

$$\le \frac{4(\mathcal{L}(\boldsymbol{\theta}_1; \mathcal{B}) - \mathcal{L}(\boldsymbol{\theta}_*; \mathcal{B}))}{T\beta_\ell\eta} + \frac{8\eta L_\infty\left(\text{tr}(\boldsymbol{\Sigma}_t) + \beta_u\right)}{T\beta_\ell}\sigma^2 + \mathcal{O}(\mu^2)$$

$$= \frac{32 L_\infty\left(\text{tr}(\boldsymbol{\Sigma}_t) + \beta_u\right)(\mathcal{L}(\boldsymbol{\theta}_1; \mathcal{B}) - \mathcal{L}(\boldsymbol{\theta}_*; \mathcal{B}))}{\sqrt{T}\beta_\ell} + \frac{\sigma^2}{T^{3/2}\beta_\ell} + \mathcal{O}\left(\mu^2\right),$$

where the first inequality is based on the assumption 3, and $\eta$ selected as $\frac{1}{8\sqrt{T}L_\infty(\max_t(\text{tr}(\boldsymbol{\Sigma}_t) + \beta_u)}$.

$\square$

