# OpenReview forum: "Memory Efficient Block Coordinate Descent Method for Forward-Only Second-Order Finetuning of LLM Models"
_ICML.cc/2025/Conference — Submitted to ICML 2025_

### Official Review · Reviewer_Nphh · 2025-03-13

**Overall Recommendation:** 2

**Summary:**

This paper proposes a memory-efficient optimization method for fine-tuning large language models by integrating a block coordinate descent scheme with Hessian-informed zeroth-order optimization. The authors claim that their method achieves reduced memory overhead while maintaining comparable accuracy to existing techniques, particularly in memory-constrained environments. Experiments are conducted on OPT-1.3B and LLaMA-2-7B.

**Claims And Evidence:**

Weak Theoretical Justification: While the paper includes some theoretical analysis adapted from prior work, it lacks rigorous new theoretical contributions that would convincingly support the efficiency and convergence claims of the proposed method.

Missing Baselines: The comparisons are limited to zeroth-order methods, while omitting strong first-order alternatives like low-memory gradient-based fine-tuning methods. Without these, the significance of the claimed efficiency improvements remains unclear.

**Essential References Not Discussed:**

The comparison is primarily focused on zeroth-order methods, ignoring strong first-order baselines like gradient-checkpointed optimizers, low-memory fine-tuning techniques (e.g., LOMO, GaLORE), and better-engineered BCD implementations.

**Experimental Designs Or Analyses:**

I check the soundness/validity of all experimental designs or analyses.

**Methods And Evaluation Criteria:**

It makes sense.

**Other Comments Or Suggestions:**

Please refer to my previous comments.

**Other Strengths And Weaknesses:**

**Strengths:**

1. The problem of memory-efficient fine-tuning for LLMs is important, especially for resource-constrained environments.

**Weaknesses:**

1. Limited Novelty: The paper primarily combines existing techniques—HiZOO and BCD—without introducing fundamentally new theoretical insights. The contribution is incremental, as the method is an adaptation rather than a novel algorithmic breakthrough.

2. Unsubstantiated Claims on Memory Efficiency: The claim that the method is a practical, convergence-enhanced alternative to MeZO is not convincingly supported. MeZO is actually more memory-efficient than the proposed method, contradicting the core motivation of the work.

3. Minimal Performance Gains: The performance improvements over MeZO are marginal (70.2 vs. 70.0 in average score), and the convergence rate is nearly identical, as observed in prior evaluations. This raises questions about whether the added complexity of BCD is justified.

4. Weak Theoretical Justification: While the paper adapts theoretical results from HiZOO, it does not provide new theoretical contributions to support the efficiency or convergence guarantees of the proposed approach.

5. Lack of Strong Baselines: The comparison is primarily focused on zeroth-order methods, ignoring strong first-order baselines like gradient-checkpointed optimizers, low-memory fine-tuning techniques (e.g., LOMO, GaLORE), and better-engineered BCD implementations.

6. Limited Scope of Experiments: The evaluation is restricted to two models (OPT-1.3B and LLaMA-2-7B), whereas HiZOO was evaluated on much larger models (up to 66B parameters). This raises concerns about the generalizability of the approach to larger-scale LLMs.

**Questions For Authors:**

Please refer to my previous questions.

**Relation To Broader Scientific Literature:**

There is no contribution of the paper related to the broader scientific literature. This paper focus on the application aspects.

**Theoretical Claims:**

There is no theoretical claim in this paper.

---

> ### Author Rebuttal · Authors · 2025-04-01
>
> **Dear Reviewer Nphh,**
>
> We sincerely appreciate your feedback. In addressing the comments from all reviewers, we have made efforts to clarify and improve each point raised. Below, we provide a concise summary of the key changes made in response to your concerns and recommendations.
>
> ### **Weaknesses:**
>
> 1. Technical contribution: At the time of our first submission, no prior work had combined second-order Newton methods, zeroth-order optimization, and block coordinate descent. Through extensive parameter searches and experiments, we demonstrate that activating partial layers with an ascending order as the block coordinate descent form succeeded and match MeZO's performance as a second-order integrated method.
> 2. Memory Efficiency: We argue that as a second-order informed method, there must be a memory-convergence tradeoff, and we managed to reduce the impact to the minimum. **New experiment results** on OPT-30B demonstrate that our method improves speed and accuracy compared to MeZO, showing breakthroughs in efficiency (Refer to our responses to **Reviewers h68x and CqZr** ).
>
> | Method | Model/Dataset   | Batch Size | Notes              | Accuracy | Training Time |
> | ------ | --------------- | ---------- | ------------------ | -------- | ------------- |
> | B-PDF  | OPT-30B (SST-2) | 16         | 2×A100 80GB nodes | 92.89\%  | **7.5 hours**     |
> | HiZOO  | OPT-30B (SST-2) | 16         | 2×A100 80GB nodes | 90.3\%   | 20.8 hours    |
> | MeZO   | OPT-30B (SST-2) | 16         | 2×A100 80GB nodes | 90.6\%   | 13.7 hours    |
> | B-PDF  | OPT-**30B** (SST-2) | 128        | 8×A100 80GB nodes | **93.6\%**   | 9.9 hours     |
> | HiZOO  | OPT-66B (SST-2) | 16         | baseline           | 93.6\%   | -             |
> | MeZO   | OPT-66B (SST-2) | 16         | baseline           | 93.6\%   | -             |
>
> 3. Performance Gains: **We can reach 1.83x wall-clock time speedup than MeZO and 2.77x speedup than HiZOO baseline, training opt-30B on SST2.** As shown in our results on larger models (30B) as above, we achieve better accuracy and computational efficiency than MeZO. We will explore block selection strategies and perform thorough parameter searches in future work to support better baselines.
> 4. Theoretical Justification: For the BCD bound, we proposed a layer selection strategy based on adjusting bandit probabilities via block gradient norms in response to **Reviewer K578**. As a starting point, this may shed light on an efficient block selection method.
> 5. First Order Baselines: In response to **Reviewer h68x**, we show experiments comparing memory requirements and explain why first-order methods (even with BCD) are less efficient than zeroth-order methods. We attach the table as follows. Note that in low-end scenarios, first-order methods remain unsuitable for comparison with zeroth-order methods because they still require significant memory overhead. Even if optimized for consumer-level 24GB GPUs, their memory usage approaches hardware limits, easily leading to unstable training (OOM errors) and small usable batch sizes (1, 2, 4). See our baseline results in response to **h68x**, whereas our zeroth-order method could run far below memory limits (<8GB), or support much larger batchsizes (128).
>
> | Method              | Batch Size | Memory Consumption | Notes       |
> | ------------------- | ---------- | ------------------ | -------------------- |
> | GaLore-AdamW (FP32) | 1 (OOM)    | OOM                | Failed due to OOM errors even at minimal batch size. |
> | GaLore-AdamW (BF16) | 8          | 42,132 MiB         | Computational overhead. Small batch size.   |
> | LOMO                | 4          | 39,910 MiB         | Stable operation with small batch size.                            |
> | MeZO                | 128        | 35,636 MiB         | Highly memory-efficient implementation supports large batch sizes. |
> | BAdam               | 2          | 22,411 MiB            | Uses paged optimizer. Causing instability in low-end scenarios.    |
>
> Note:
> These results illustrate significant differences in memory requirements between first-order and zeroth-order methods.
> First-order approaches, even though including memory-efficient or BCD variants, still incur memory overhead from storing states such as activations. When optimizing the first layer in a first-order BCD framework, peak memory usage remains high due to the need to retain activations for gradient computation in subsequent layers.
>
> 6. Scope of Experiments: After extending the parameter scale, our SST-2 result reaches 92.8\% is close to the accuracy reported by HiZOO for 66B models (93.6\%). We added an additional experiment with 8 GPUs (16 batch size per GPU), achieving 93.6\% accuracy with the fastest speed. This could prove the generalizability of our approach to larger-scale LLMs. We attach this result to the table "Memory Efficiency".
>
> Thank you for your help in improving our work. We look forward to your feedback.

---

### Official Review · Reviewer_CqZr · 2025-03-13

**Overall Recommendation:** 3

**Summary:**

The authors propose a new zero-order optimizer for fine-tuning the pre-trained model to the downstream task that incorporates second-order information. The main issue addressed in the study is the infeasible memory consumption of classical optimizers for the fine-tuning process. The main idea is to use a block coordinate descent framework and update only a part of layers' parameters in every iteration. This approach makes the low-memory custom devices appropriate for fine-tuning the LLMs. LLaMa2-7b and OPT-1.3B models are considered for GLUE and SuperGLUE downstream tasks in experiments. The proposed approach leads to a reduction in memory footprint while preserving the same final accuracy.

**Claims And Evidence:**

Most claims in the manuscript are supported by numerical or theoretical evidence. However, I would like to see training loss and test loss for the considered tasks and the corresponding runtime in Figure 3. In addition, the stability analysis for the proposed approach is ignored. I would suggest showing the dependence of the convergence on the batch size used in the gradient estimation from the zero-order information. Using batchsize=1 provides an extremely noisy estimate, I guess.

**Essential References Not Discussed:**

Missing references to alternative zero-order methods developed for training neural networks: ReLIZO: Sample Reusable Linear Interpolation-based Zeroth-order Optimization, Xiaoxing Wang, Xiaohan Qin, Xiaokang Yang, Junchi Yan, NeurIPS 2024

**Experimental Designs Or Analyses:**

The design of the presented numerical experiments is sound and valid for the considered task. The selection of the competitors is also well-motivated and meaningful.

**Methods And Evaluation Criteria:**

The methods and evaluation criteria make sense and align with the problem stated in Section 3.

**Other Comments Or Suggestions:**

No other comments or suggestions.

**Other Strengths And Weaknesses:**

I see three main weaknesses in this submission.
1. The experiments consider only two medium/small-scale models. So, the robustness of the proposed method in fine-tuning larger models remains unclear. I am sure the memory footprint will be smaller, but will the accuracy be preserved as in the non-block strategy?
2. While many blocking strategies are discussed, the single simplest blocking strategy is tested. I am sure many natural heuristic blocking strategies require the same amount of memory and could provide better results. For example, one can update all even layers and then all odd layers or something similar.
3. This study completely ignores low-precision formats. At the same time, quantization is the natural competitor in reducing the memory footprint during fine-tuning. The larger models in BF16 or even lower-bit formats could be discussed and tested on the mentioned GPUs. The synergy of such a memory-efficient optimizer and low-bit formats could provide more opportunities for fine-tuning huge models in user-level devices.

**Questions For Authors:**

1. What quantities are presented in Table 3?
2. Why do authors exclude BAdam from the competitors? Its performance could highlight the impact of the inexactness in the gradient estimation with the approximate second-order information.
3. How much the quality degrades if one uses the first-order approximation of the gradient based on $L(\theta)$ and $L(\theta+ \delta)$?

**Relation To Broader Scientific Literature:**

The key contribution of the submitted manuscript is the combination of the zero-order optimization procedure, second-order preconditioner, and block coordinate descent framework. The authors find a practically important setup where such a combination becomes crucial for the overall performance of the fine-tuning process.

**Theoretical Claims:**

The manuscript does not provide any rigorous proof, only the result of modification for the HiZOO theorem. A convergence rate like $O(1/ \sqrt{T})$ looks reasonable, although I did not check the proof line-by-line.

---

> ### Author Rebuttal · Authors · 2025-04-01
>
> **Dear Reviewer CqZr,**
>
> Thank you for your thoughtful questions and insights.
> We appreciate your feedback and will address it as follows:
>
> ### **Claims and Evidences:**
>
> Training and test loss for figure 3, train loss are smoothed as in the figure:
> | Method  | Training Loss | Test Loss |
> |---------|--------------:|----------:|
> | HiZOO   |        0.1973 |    0.2410 |
> | B-PDF   |        0.2040 |    0.2266 |
> | MeZO    |        0.2308 |    0.2410 |
>
> Stability analysis:
> To address your request, we test under extreme conditions with batch size = 1 (without extensive tuning), observing a accuracy drop: SST2 acc=0.8337 (baseline 0.917).
> Since using batch size = 1 introduces significant variance, in practice, we have avoided this by employing batch size $\ge$ 16, which also easily fits within memory limits for zeroth-order methods (see also our response to **Reviewer h68x**, Experimental Design).
>
> Theoretical Claims: We direct you to the randomized BCD framework discussed in our response to **Reviewer K578**.
>
> ### **Missing Reference:**
> We appreciate your suggestion to include ReLIZO (Wang et al., NeurIPS 2024).
> We note that this work improves computation efficiency by modeling the gradient estimation into a QCLP problem, and applying query-reuse strategy in zeroth-order optimization, achieving impressive results across tasks (especially, 93.4\% accuracy on SST-2).
> We will cite it as a representative example of computationally efficient zeroth-order techniques.
>
> ### **Weaknesses:**
>
> 1. As detailed in our response to Reviewer h68x, we **expanded our experiments** training OPT-30B on SST2 task, using 2×A100 nodes.
> B-PDF achieves 92.89\% accuracy in 7.5 hours, outperforming HiZOO (90.3\%, 19.8 hours) and MeZO (90.6\%, 13.7 hours). This result validates the scalability and efficiency of our method when scaling up.
>
> | Method | Model/Dataset   | Batch Size | Hardware           | Accuracy | Training Time |
> | ------ | --------------- | ---------- | ------------------ | -------- | ------------- |
> | B-PDF  | OPT-30B (SST-2) | 16 | 2×A100 80GB nodes | 92.89\%  | 7.5 hours     |
> | HiZOO  | OPT-30B (SST-2) | 16 | 2×A100 80GB nodes | 90.3\%   | 20.8 hours    |
> | MeZO   | OPT-30B (SST-2) | 16 | 2×A100 80GB nodes | 90.6\%   | 13.7 hours    |
>
> 2. We tested some additional strategies: We evaluated following block selection strategies on OPT-1.3B, SST-2 (We note that arameters are not fully searched in these experiments, so there is room for improvement):
>
> | Strategy                                     | Accuracy |
> |-------------------------------------|----------|
> | Ascending Order (ours default)  | 91.9%    |
> | Gauss-Southwell-Diagonal        | 90.71%   |
> | Random Reshuffling                  | 90.71%   |
> | An Odd-Even Staged Strategy  | 91.40%   |
>
> *Note: In practice, we found counting gradient norm introduces heavy computational overhead, making it less efficient than the natural ascending order or random sampling. This matches the results in BAdam [4]. As for the odd-even staged strategy, we perform [1, 3, 5, 7], followed by [2, 4, 6, 8], ... , as integrating our active block sets with your suggestion.*
>
> > [4] Luo, Qi, Hengxu Yu and Xiao Li. “BAdam: A Memory Efficient Full Parameter Optimization Method for Large Language Models.” *Neural Information Processing Systems* (2024).
>
> We will add this analysis to appendix and extend it in future work. Currently, we see that the ascending order remains optimal for balancing performance and simplicity.
>
> 3. Regarding the application prospects of combining low-precision formats with memory-efficient optimizers, we fully agree with your point of view. In our current work, we conduct experiments based on MeZO framework with FP16 precision, and in the future we could explore combining our method and lower precision training. In addition, we observed challenges in utilizing Hessian information matrices under low-precision conditions during our experiments, and have applied clamping methods to stabilize numerical computations. We are following advancements in recent work and advanced GPU architectures supporting FP8 quantization, which we believe that quantization methods will significantly enhance training efficiency in the future.
>
> ### **Questions:**
>
> 1: Quantities in Table 3: The quantities represent test accuracy.
>
> 2: Due to low-end system instability on our consumer-grade hardware (RTX 4090 on an old and low-bandwidth motherboard), BAdam caused overheating and crashes.
>
> For comparison, we have tested Adamw-HF(HuggingFace version) accuracy on SST-2, opt-1.3b: 93.70\% (with BS=8 LR=5e-6, peak allocated memory=10450MB, 10k training steps).
>
> 3: We conduct an experiment under a MeZO setting, which shows that the accuracy slightly improves (91.7\% to 91.86\%) when using single perturbations.
>
> (MODEL=opt-1.3b TASK=SST2 BS=16 MODE=ft LR=3e-7 EPS=1e-3 STEPS=20000, running time is 2h57min on an A6000 node)
>
> We sincerely express our gratitude for your guidance in improving our work.

---

### Official Review · Reviewer_h68x · 2025-03-14

**Overall Recommendation:** 3

**Summary:**

This paper proposes B-PDF, a memory efficient bcd-newton optimization method for LLM fine-tuning, especially for low-end devices, which integrates block coordinate descent with a zeroth-order Newton-method optimizer. This approach reduces memory overhead by updating parameters and diagonal Hessian information in a layer-wise BCD scheme. Experiments show that the proposed method reduces the memory-intensive bottleneck of the second-order optimization while maintaining performance.

**Claims And Evidence:**

1. memory efficiency: experiments show that B-PDF reduces memory cost comparing to hizoo on opt-1.3B and llama2-7B.
2. convergence rate: Figure 3 shows that B-PDF converges better than mezo, and matches hizoo’s accuracy with faster wall-clock speed.
3. practical utility: B-PDF can fine-tune llama2-7b on an RTX A6000, a practical case for relatively low-resource deployment.

**Essential References Not Discussed:**

These methods could be included as baselines to further prove performance:
1. GaLore: Memory-Efficient LLM Training by Gradient Low-Rank Projection
2. zo-adam: Revisiting Zeroth-Order Optimization for Memory-Efficient LLM Fine-Tuning: A Benchmark

**Experimental Designs Or Analyses:**

1.measurements on opt-1.3B/llama2-7B are valid.
2.baselines: missing comparisons with optimizers such as GaLore, which are relevant for memory efficiency and first-order optimization.

**Methods And Evaluation Criteria:**

1. Methods: BCD with Hessian-informed ZO is well motivated to reduce the memory cost and boost convergence, addressing the memory-convergence trade-off via layer-wise updates.
2. Evaluation: GLUE benchmarks are standard, and the low-end settings using small batch sizes seems practical for consumer-level gpus in real-world scenarios.

**Other Comments Or Suggestions:**

N/A

**Other Strengths And Weaknesses:**

Strengths:

1.novelty: the problem and background is well motivated. The integration of BCD with Hessian-informed zeroth-order optimization is a sensible contribution to memory-efficient LLM fine-tuning, which avoids backpropagation and reduces storage.
2. technical impact: the proposed method is creative, resolving a key bottleneck in second-order methods, it sure can benefit real-world use case.
3. The paper is well-written and easy to follow.

Weaknesses:

1.baseline: while comparisons with hizoo and mezo are thorough, including bcd-mezo and recent memory-efficient methods (e.g. galore) would better prove B-PDF’s performance.
2. scalability: are there testing results on models >7B?

**Questions For Authors:**

1. Some efficient baselines like GaLore, bcd-mezo and ZO-Adam is not included as a baseline. Could B-PDF save more memory than them?
2. How does block selection impact performance?

**Relation To Broader Scientific Literature:**

Findings: the work rethink the memory overhead and show the memory-convergence tradeoff in previous algorithms (hizoo).
Ideas: the work bridges gaps in zeroth-order optimization (mezo, hizoo) and block-wise training (badam, lisa).
Results: the work enables full-parameter fine-tuning with memory efficiency comparable to mezo and lora-like PEFT methods. it demonstrates that second-order brings faster convergence without sacrificing memory efficiency, and show practical 1.3B~7B model adaptation on relatively low-performance GPUs (4090 and A6000).

**Theoretical Claims:**

The convergence proof aligns with the paper’s focus.

---

> ### Author Rebuttal · Authors · 2025-04-01
>
> **Dear Reviewer h68x,**
>
> Thanks for your feedback and valuable suggestions.
> Below, we address each of your concerns to improve our manuscript.
>
> ### **Experimental Designs and Question 1:**
>
> Baseline Comparisons with GaLore and Other Methods:
> We have carefully considered your comments and those from Reviewers CqZr and Nphh, regarding comparisons with first-order memory-efficient baselines such as BAdam, GaLore, and LOMO.
>
> Below, **new experiments** demonstrate the high memory consumption of these methods, conducted on a single RTX A6000 (48GB) with the LLaMA-3-8B model loaded in FP32 format for the SST-2 task (table shown in 5.First Order Baselines to Reviewer **Nphh**) :
>
> - GaLore-AdamW: Encountered out-of-memory (OOM) errors even with a batch size of 1. When loaded in BF16 format, it could run with a maximum batch size of 8, consuming 42,132 MiB of memory. However, the initial training steps were significantly slower due to its additional computational overhead.
> - LOMO: Achieved a maximum batch size of 4 with 39,910 MiB memory usage.
> - MeZO: Supported a maximum batch size of 128 with 35,636 MiB memory consumption.
> - BAdam: Required 23.5 GB of memory with a batch size of 2, as reported in the paper [1]. We note that while the official implementation utilizes a paged optimizer to reduce memory pressure, we observed high I/O costs between CPU and GPU memory in our test environment (consumer-grade motherboard and RAM), leading to system instability. This limitation is less pronounced in data center environments with optimized cooling and hardware support, but it highlights challenges for low-end systems.
>
> > [1] Luo, Qi, Hengxu Yu and Xiao Li. “BAdam: A Memory Efficient Full Parameter Optimization Method for Large Language Models.” Neural Information Processing Systems (2024).
>
> These results illustrate significant differences in memory requirements between first-order and zeroth-order methods.
> First-order approaches, including memory-efficient or BCD variants, still incur memory overhead from storing activations for backpropagation.
> For example, when optimizing the first layer in a first-order BCD framework, peak memory usage remains high due to the need to retain activations for gradient computation in subsequent layers.
> Thus, our original experiments focused on zeroth-order baselines. We will attach the comparison in the supplementary material to enable a more comprehensive understanding of the memory efficiency of first-order methods.
>
> Regarding **ZO-Adam**, while it demonstrates higher accuracy than MeZO in certain scenarios, its memory footprint is substantially larger. As reported by Zhang et al. [2], fine-tuning the full OPT-13B model on the MultiRC dataset with a batch size of 4 requires 64 GB for ZO-SGD and 158 GB for ZO-Adam, exceeding the capacity of typical low-end devices.
>
> > [2] Zhang, Yihua, Pingzhi Li, Junyuan Hong, Jiaxiang Li, Yimeng Zhang, Wenqing Zheng, Pin-Yu Chen, Jason D. Lee, Wotao Yin, Mingyi Hong, Zhangyang Wang, Sijia Liu and Tianlong Chen. “Revisiting Zeroth-Order Optimization for Memory-Efficient LLM Fine-Tuning: A Benchmark.” ArXiv abs/2402.11592 (2024).
>
> To ensure comprehensive analysis, we will include detailed comparisons with these methods in the supplementary material. This addition will clarify the trade-offs between memory efficiency and computational requirements across different optimization paradigms.
>
> ### **Weaknesses:**
>
> Scalability to Models $>$ 7B Parameters:
>
> While our initial experiments focused on OPT-1.3B and LLaMA-2-7B due to hardware limitations, now we have conducted **an additional experiment** to address scalability concerns and further validate our method. For OPT-30B fine-tuned on the SST-2 dataset with a batch size of 16 (2xA100 80GB nodes),
>
> **B-PDF** achieved 92.89\% accuracy in  **7.5 hours** , significantly outperforming HiZOO (90.3\% accuracy [3], 20.8 hours) and MeZO (90.6\% accuracy [3], 13.7 hours).
>
> The improved accuracy and reduced runtime highlight the efficiency of our proposed method for both low-end and high-end environments. While zeroth-order methods traditionally trade accuracy for memory savings, and second order integration brings significant computing overhead, our method mitigates this trade-off, enabling competitive performance even on larger models.
>
> > [3] Zhao, Yanjun, Sizhe Dang, Haishan Ye, Guang Dai, Yi Qian and Ivor Wai-Hung Tsang. “Second-Order Fine-Tuning without Pain for LLMs: A Hessian Informed Zeroth-Order Optimizer.” ArXiv abs/2402.15173 (2024).
>
> ### **Question 2:**
>
> Impact of Block Selection Strategies:
>
> Due to length limits, please refer to our answer to **Reviewer CqZr** for a detailed discussion on the impact of block selection strategies. We will include this analysis in the supplementary material.
>
> We appreciate your guidance in helping us strengthen this work.

---

### Official Review · Reviewer_K578 · 2025-03-14

**Overall Recommendation:** 2

**Summary:**

The paper proposes a zero-order method for fine-tuning large language models (LLMs), utilizing a block coordinate descent approach to reduce memory costs.
In this approach, blocks are defined as layers of the LLM, which are updated individually while the remaining layers are frozen.
To improve convergence speed, the authors incorporate Hessian information, which is estimated in a forward-only manner.
The method's competitiveness is demonstrated by comparing it against other zero-order methods, namely in terms of memory usage and time efficiency.

**Claims And Evidence:**

The authors are not entirely honest when claiming an "improved block coordinate descent scheme" as their contribution.
In practice, Column 1 Lines 282-292, they use a standard BCD method with an ascending order rule.

**Essential References Not Discussed:**

None

**Experimental Designs Or Analyses:**

The experimental design is sound.
In addition the authors provide baseline comparison with first-order methods.

**Methods And Evaluation Criteria:**

The evaluation is fair
- the authors assess the timing gains and memory savings relative to other zero-order methods
- the accuracy of the method is also being evaluated

**Other Comments Or Suggestions:**

None

**Other Strengths And Weaknesses:**

- Fix line numbers in Algorithm 1
- Avoid bold statement such as the one in Column 2 Line 295 "The consistent use of random vectors and selective parameter perturbation further enhance the method’s memory efficiency."
- in figure 2, color code in the bar chart is confusing especially between "parameters" and "gradients" bins

**Questions For Authors:**

No further questions

**Relation To Broader Scientific Literature:**

From the experimental results, the method provides only minor improvements compared to MeZO, as seen in Table 3.
Additionally, the claim of improving convergence speed is modest, with a maximum improvement of only 7%, according to Table 2.

**Theoretical Claims:**

Equation (2) does not correctly transcribe the update from [1].
Specifically, there is an issue with the second term on the right-hand side, which causes it to diverge from the standard EMA formulation.
In EMA, the first parameter corresponds to an accumulation of past updates, however the meaning of the absolute value $| \Sigma_t |$ is undefined in the equation.

There are concerns with Algorithm 1
- There is confusion in the indices $i, s, t$ which makes the algorithm unclear.
- The projected gradient steps should be computed with the Hessian approximation before the update, but this is not done.
- For the weight updates, the same random direction z used to compute the perturbation should be used for the updates, but there is no hint about that in the algorithm.
- The algorithm loops over $\theta_i$ in $\theta_b$, which gives the impression that a batch of blocks is being updated, but this is not clarified.
- The EMA step in line 17 does not correspond to an EMA process as described in the literature.

The convergence proof in Appendix D is invalid. [1], on which the authors base their analysis, provides a proof for the whole-update method, while the authors apply this analysis to BCD.
The proof sketch for full updates and coordinate updates are different, and BCD can lead to cyclic behavior, preventing convergence (see [3], Example 3.1).
Additionally, the role of the blocks is not addressed in the proof, which make it irrelevant to the specific case of BCD.


---
.. [1] Zhao, Yanjun, et al. "Second-order fine-tuning without pain for llms: A hessian informed zeroth-order optimizer." arXiv preprint arXiv:2402.15173 (2024).

.. [2] Tarvainen, Antti, and Harri Valpola. "Mean teachers are better role models: Weight-averaged consistency targets improve semi-supervised deep learning results." Advances in neural information processing systems 30 (2017).

.. [3] Wright, Stephen J. "Coordinate descent algorithms." Mathematical programming 151.1 (2015): 3-34.

---

> ### Author Rebuttal · Authors · 2025-04-01
>
> **Dear Reviewer K578,**
>
> We sincerely appreciate your thoughtful review and apologize for the confusion caused by the typos and unclear claims. We are grateful for your careful reading and will address each point thoroughly.
>
> ### **Theoretical Claims:**
>
> For Equation 2 and the EMA, the first term $\Sigma_{t+1}^{-1}$ represents the stored Hessian.
> We will add a hat to indicate the $\hat\Sigma_t$ is the estimated Hessian, as described in line 311 of Algorithm 1, and the use of the absolute value sign to ensure non-negativity. We will remove the term diag() in line 312 since it is already in diagonal form. This revision will address your concern about the unaligned EMA process. (point 5)
>
> Regarding the confusion in Algorithm 1:
>
> - point 1, indices: we will explain them more carefully as below.
> - point 4, the index $\pi_b$ means that in all we have $b$ active blocks for the step.
>   We find that selecting more than one blocks for efficiency works well, and this could be a hyperparameter to search for. In practice, using 4 blocks works effectively.
> - The parameter $\mu$ represents the in-place perturbation, and we perform forward passes for $\theta$, $\theta+\mu\Sigma^{\frac{1}{2}}$, and $\theta-\mu\Sigma^{\frac{1}{2}}$ with $\mu_i$ in the directions 0, +$\mu$, and -2$\mu$.
> - point 3, The index $s$ refers to the seed, and we use it to sample the same random direction $z$ for both perturbation and updates, which is a key idea in MeZO.
> - point 2,  we admit that our implementation follows HiZOO's design in computing Hessian first. In our practice, block switching is frequent, and this implementation sometimes converges faster since the Hessian initialization is an identity matrix. We will try to ablate its impact in revisions.
>
> For the Convergence Proof for BCD, as appropriately noted, the process  different from full-parameter optimization can raise theoretical concerns.
> To address this, we provide the analysis to a randomized BCD framework by incorporating a probabilistic block selection mechanism.
> Below is the proof sketch:
>
> For brevity, consider the objective function $\mathcal{L}(\theta)$, where the parameters $\theta = [\theta_1, \dots, \theta_D]$ are partitioned into $D$ blocks. In each iteration, a block $i \in \\{1,\dots,D\\}$ is randomly selected with probability $p_i$ to be updated.
> At each iteration, the gradient block $ \hat{\nabla} \mathcal{L}\_{t,i} $ is retained with probability $ p_{t,i} $ and dropped otherwise. This sparsification is formalized as:  $\hat{\nabla} \mathcal{L}(\theta_t) = \sum_{i=1}^D \frac{\hat{\nabla} \mathcal{L}\_{t,i}}{p_{t,i}} Z_{t,i},$ where $ Z_{t,i} \sim \text{Bernoulli}(p_{t,i}) $. The sparsified gradient $\hat{\nabla} \mathcal{L}(\theta_t)$ is unbiased.
> Under following assumptions:
> - block L-smoothness $||\nabla_i \mathcal{L}(\theta) - \nabla_i \mathcal{L}(\theta')|| \leq L_i ||\theta_i - \theta_i'||$,
> - bounded gradient estimation variance $\mathbb{E}[|| \hat{\nabla}\_i \mathcal{L}(\theta) - {\nabla}\_i \mathcal{L}(\theta)||^2]\le \sigma_i^2$,
> - Hessian preconditioning constraints  $ 0 < \beta_\ell \leq \lambda_{\min}(\Sigma_i) \leq \lambda_{\max}(\Sigma_i) \leq \beta_u $,
> - Bregman divergence bound: $B(\theta, \theta') \leq R^2 $ for all $ \theta, \theta'$.
> Define $\Sigma = \text{diag}(p_1 \Sigma_1, \dots, p_D \Sigma_D),$
>
> we can finally prove that,
>
> $\mathbb{E} \sum_{t=1}^T \sum_{i=1}^D\left[||\nabla \mathcal{L}(\theta_d^t)||^2_{\Sigma}\right] \le \frac{2 R^2}{\eta}+16\eta L_\infty (\text{tr}(\Sigma_\infty)+\beta_u) \sum_{t=1}^T \sum_{i=1}^D \left( \frac{||\nabla_i \mathcal{L}(\theta^t)||^2_{\Sigma_i}}{p_{i,t}}+\sigma_i^2 \right)$
>
> Fortunately, to the best of our knowledge, existing approach can solve  $p_{t,i} = \frac{||\hat{\nabla} \mathcal{L}\_{t,i}||}{\sum_{d=1}^D ||\hat{\nabla} \mathcal{L}\_{t,d}||}$ with a bandit trick, and it does not affect the $\mathcal{O}(1/\sqrt{T})$ bound[1].
>
> > [1] Communication-efficient Distributed Learning for Large Batch Optimization. Rui Liu, Barzan Mozafari, PMLR 162:13925-13946, 2022.
>
> Thus, we extend the convergence analysis to a block coordinate descent setting, demonstrating the applicability of our method. This also matches with that, while BCD may prevent convergence in non-convex settings, our experiments demonstrate stable convergence in practice. Due to response length limits, we can discuss this further in the final response period and we welcome your suggestions.
>
> ### **Broader Scientific Literature:**
>
> Regarding performance: Initial experiments were limited by hardware.  However, with additional computational resources, we have benchmarked on OPT-30B to validate scalability, which achieves better accuracy with less training time for larger models. Please, refer to our response to **Reviewers h68x and Nphh**. We leave hyperparameter tuning and BCD schemes for future work.
>
> Thank you for your constructive feedback on strengthening both our theory and practical validation.

---

> > ### Comment · Reviewer_K578 · 2025-04-02
> >
> > Upon reading the answer, here is my insight:
> >
> > The authors do not fully address my theoretical concerns, making the response difficult to assess.
> > - The theoretical results of the paper remain questionable. Providing a complete proof separately would have been better
> > - Similarly, a revised version of the algorithm would have been better

---

> > > ### Author Response · Authors · 2025-04-07
> > >
> > > Dear Reviewer K578,
> > >
> > > Thanks for your patience. With the expanded space,  we now provide a more comprehensive analysis of the randomized version of our BCD algorithm.
> > >
> > > Considering all necessary assumptions from [1], [2].
> > > Suppose the model has $D$ blocks.
> > > Denote $\theta_{t,[i]} = [0,\dots,\theta_{t,i},\dots,0]$ and $g_{t,[i]} = [0,\dots,g_{t,i},\dots,0]$.
> > > At the $t$-th iteration, the $i$-th block in BCD has parameter $\theta_{t,[i]}$.
> > > The bracket notation $[\,]$ also applies to the diagonal Hessian $\Sigma_t$ and random perturbation $u_t$.
> > >
> > > Each block has a probability $p_i$ of being selected. At the $t$-th iteration, the sampling indicator $Z_{t,i}$ is drawn from a Bernoulli distribution with parameter $p_i$. We use the gradient estimate:
> > > \begin{equation}
> > > \tilde{g}\_{t,[i]} = \frac{Z_{t,i}}{p_i} \cdot \frac{L\left(\theta_t + \sum_{d=1}^D \mu\Sigma_{t,[d]}^{1/2}u_{[d]}\right) - L\left(\theta_t - \sum_{d=1}^D \mu\Sigma_{t,[d]}^{1/2}u_{[d]}\right)}{2\mu} \cdot \Sigma_{t,[i]}^{1/2}u_{[i]},
> > > \end{equation}
> > > where the term $\frac{Z_{t,i}}{p_i}$ ensures unbiased estimation.
> > >
> > > From Taylor expansion:
> > > \begin{equation}
> > > \Delta L = 2\mu\nabla^\top L(\theta_t)\sum_{d=1}^D \Sigma_{t,[d]}^{1/2}u_{[d]} + \mathcal{O}(\mu^2).
> > > \end{equation}
> > >
> > > The expectation yields:
> > >
> > > \begin{aligned}
> > > \tilde{g}\_{t,[i]} &= \frac{Z_{t,i}}{p_i}\sum_{d=1}^D \Sigma_{t,[i]}^{1/2}u_{[i]}u_{[d]}^\top\Sigma_{t,[d]}^{1/2}\nabla L(\theta_t) + \mathcal{O}(\mu), \\\\
> > > \mathbb{E}[\tilde{g}\_{t,[i]}] &= \Sigma_{t,[i]}\nabla L(\theta_t) + \mathcal{O}(\mu).
> > > \end{aligned}
> > >
> > > The update rule is given by:
> > > \begin{equation}
> > > \theta_{t+1,[i]} = \theta_{t,[i]} - \eta_t \cdot \tilde{g}\_{t,[i]}.
> > > \end{equation}
> > >
> > > Under the block Lipschitz assumption:
> > > \begin{aligned}
> > > L(\theta_{t+1}) - L(\theta_t)
> > > &\leq \sum_{i=1}^D \left\langle \nabla L(\theta_t), \theta_{t+1,[i]} - \theta_{t,[i]} \right\rangle + \frac{L_\infty}{2}\|\theta_{t+1} - \theta_t\|^2 \nonumber \\\\
> > > &= -\eta_t \sum_{i=1}^D \left\langle \nabla L(\theta_t), \tilde{g}\_{t,[i]} \right\rangle + \frac{L_\infty\eta_t^2}{2}\|\tilde{g}\_t\|^2.
> > > \end{aligned}
> > >
> > > Take expectation and according to HiZOO proof [2]:
> > > \begin{aligned}
> > > \mathbb{E}[L(\theta_{t+1})] - \mathbb{E}[L(\theta_t)]
> > > &\leq -\eta_t||\nabla L(\theta_t)||\_{\Sigma_t}^2 + \eta_t\mathcal{O}(\mu||\nabla L(\theta_t)||) + \frac{L_\infty\eta_t^2}{2}\mathbb{E}||\tilde{g}\_t||^2 \\\\
> > > &\leq -\frac{\eta_t}{2}||\nabla L(\theta_t)||\_{\Sigma_t}^2 + \frac{L_\infty\eta_t^2}{2}\mathbb{E}||\tilde{g}\_t||^2.
> > > \end{aligned}
> > >
> > > Summing over $T$ iterations:
> > > \begin{aligned}
> > > \sum_{t=1}^T \frac{\eta_t}{2}||\nabla L(\theta_t)||\_{\Sigma_t}^2
> > > &\leq \mathbb{E}[L(\theta_1)] - \mathbb{E}[L(\theta^*)] + \frac{L_\infty\eta_t^2}{2}\sum_{t=1}^T\sum_{i=1}^D \frac{||g_{t,[i]}||^2}{p_i}.
> > > \end{aligned}
> > >
> > > The optimal probabilities minimizing the second term are:
> > > \begin{equation}
> > > p_{t,i} = \frac{||g_{t,[i]}||}{\sum_{i=1}^D ||g_{t,[i]}||}, \quad \forall d.
> > > \end{equation}
> > >
> > > JointSpar[1] solves this via bandit optimization and achieves $\mathcal{O}\left(\frac{1}{\sqrt{T}}\right)$ convergence rate.
> > >
> > > > [1] Communication-efficient Distributed Learning for Large Batch Optimization. Rui Liu, Barzan Mozafari, PMLR 162:13925-13946, 2022.
> > >
> > > > [2] Zhao, Yanjun, et al. "Second-order fine-tuning without pain for llms: A hessian informed zeroth-order optimizer." arXiv preprint arXiv:2402.15173 (2024).
> > >
> > > An improved algorithm version is as below:
> > >
> > > **Algorithm 1** Training Pipeline of the Proposed B-PDF
> > > **Input**: Parameters $\theta \in \mathbb{R}^d$, loss function $\mathcal{L}$, perturbation scale $\mu$, learning rate $\eta$, smooth scale $\alpha$
> > > **for** $t = 1, \dots, T$ **do**
> > >  **1.** Select block $\theta_{\pi_b} \in \theta$ according to the BCD rule
> > >  **2.** **if** a new block is selected **then**
> > >   $\Sigma_1 \gets \mathbf{I}\_{|\theta_{\pi_b}|}$ *// Diagonal Hessian initialization*
> > >  **3.** Freeze other layers
> > >  **4.** Sample a random seed $s$ *// First-time sampling*
> > >  **5.** **for** $\mu_i = 0, +\mu, -2\mu$ **do**
> > >   **6.** **for** $\theta_i \in \theta_{\pi_b}$ **do**
> > >     Sample $z\_i \sim \mathcal{N}\_s(0, \mathbf{I}\_{|\theta_i|})$
> > >     $\theta_i \gets \theta_i + \mu_i \Sigma^{1/2}\_{t,i} z\_i$ *// In-place perturbation*
> > >       **end for**
> > >    $\ell_{\texttt{sign}(\mu_i)} \gets \mathcal{L}(\theta)$  *// forward (3x in interations)*
> > >    **end for**
> > >  **7.** Compute projected gradient:
> > >   projected\_grad $\gets (\ell_{+} - \ell_{-}) \Sigma^{1/2}\_t / 2\mu$
> > >  **8.** Update Hessian:
> > >   $\hat{\Sigma}\_{t+1} \gets \frac{\Delta \ell}{2 \mu^2} \Sigma^{-1/2}\_{t} z z^\top \Sigma^{-1/2}\_{t}$
> > >  **9.** Smooth covariance:
> > >   $\Sigma_{t+1}^{-1} \gets (1 - \alpha_t) \Sigma_{t}^{-1} + \alpha_t \left| \hat{\Sigma}\_{t+1} \right|$
> > >  **10.** Reset random number generator with seed $s$
> > >  **11.** **for** $\theta_i \in \theta_{\pi_b}$ **do**
> > >   Sample $z\_i \sim \mathcal{N}\_s(0, \mathbf{I}_{|\theta_i|})$
> > >   $\theta_i \gets \theta_i - \eta_t \cdot$ projected\_grad $\cdot z_i$
> > >  **12.** **end for**
> > > **end for**

---

### Decision · Program_Chairs · 2025-05-01

**Decision:**

Reject

**Comment:**

This paper proposes a memory-efficient fine-tuning approach for LLMs using a forward-only, Hessian-informed zeroth-order optimizer with block coordinate descent (BCD). The motivation is sound, and the method demonstrates some empirical gains on medium-sized models. However, the reviewers raised several concerns that have not been fully addressed by the authors: (1) the novelty is limited, as BCD is applied in a standard way with little innovation; (2) the theoretical analysis is insufficient to justify the convergence claim; and (3) the empirical improvements are marginal, and more experiments are needed. Given these concerns, I recommend rejection.